# Properties of a Single Amino Acid Residue in the Third Transmembrane Domain Determine the Kinetics of Ambient Light-Sensitive Channelrhodopsin

**DOI:** 10.3390/ijms24055054

**Published:** 2023-03-06

**Authors:** Akito Hatakeyama, Eriko Sugano, Tatsuki Sayama, Yoshito Watanabe, Tomoya Suzuki, Kitako Tabata, Yuka Endo, Tetsuya Sakajiri, Tomokazu Fukuda, Taku Ozaki, Hiroshi Tomita

**Affiliations:** Laboratory of Visual Neuroscience, Graduate Course in Biological Sciences, Division of Science and Engineering, Iwate University, 4-3-5 Ueda, Morioka 020-8551, Iwate, Japan

**Keywords:** channelrhodopsin, channel kinetics, retina, photoreceptor degeneration

## Abstract

Channelrhodopsins have been utilized in gene therapy to restore vision in patients with retinitis pigmentosa and their channel kinetics are an important factor to consider in such applications. We investigated the channel kinetics of ComV1 variants with different amino acid residues at the 172nd position. Patch clamp methods were used to record the photocurrents induced by stimuli from diodes in HEK293 cells transfected with plasmid vectors. The channel kinetics (τon and τoff) were considerably altered by the replacement of the 172nd amino acid and was dependent on the amino acid characteristics. The size of amino acids at this position correlated with τon and decay, whereas the solubility correlated with τon and τoff. Molecular dynamic simulation indicated that the ion tunnel constructed by H172, E121, and R306 widened due to H172A variant, whereas the interaction between A172 and the surrounding amino acids weakened compared with H172. The bottleneck radius of the ion gate constructed with the 172nd amino acid affected the photocurrent and channel kinetics. The 172nd amino acid in ComV1 is a key residue for determining channel kinetics as its properties alter the radius of the ion gate. Our findings can be used to improve the channel kinetics of channelrhodopsins.

## 1. Introduction

Photoreceptor degeneration caused by diseases, such as retinitis pigmentosa (RP) and age-related macular degeneration, leads to complete blindness following reduced visual acuity or night blindness [1]. Several approaches have been investigated to compensate for the loss of function of photoreceptors, such as retinal prosthesis and stem cell transplant; however, useful therapeutic strategies have not been established. Recently, gene therapies using various optogenetic genes to target retinal ganglion cells (RGCs) have been developed for restoring vision in patients with RP. This approach would be useful for patients with RP because the effect of the treatment is not influenced by the causative gene of RP. The feasibility of this strategy has been demonstrated in animal models in functional restoration [2,3,4,5] and safety studies [6,7,8,9,10,11] using channelrhodopsins (ChRs). Some of these gene therapies are also being evaluated in clinical trials involving patients with RP (NCT02556736 and NCT03326336). Indeed, partial recovery of visual function in a blind patient after optogenetic therapy was reported recently [12].

Different types of ChRs have been identified from various species, such as *Chlamydomonas reinhardtii* [13,14,15,16], *Volvox carteri* [17], *Chloromonas oogama* [18], and *Guillardia theta* [19]. In addition, functional improvements have been attempted via the construction of chimeric proteins [20,21,22] or replacement of specific amino acids [23,24,25]. In gene therapies, the recovered function depends on the transduced gene itself. For example, ChR2, which is mainly sensitive to blue light, is being investigated in clinical trials for the treatment of patients with RP; it is considered that the restored vision in patients who received ChR2 gene therapy is limited to wavelength sensitivity. Therefore, ChRs that respond to light of a wide range of wavelengths have been developed; they include mVChR1 [21] and ChrimsonR [18]. However, the threshold of light intensity for activating ChRs is approximately in the order of mW/mm^2^ [18,26]. Although ChRs could be a potential treatment for restoring vision, the sensitivity of ChRs to the intensity of light that we are exposed to in daily life is too low.

Improvement of light sensitivity of ChRs has been investigated using various approaches. An increase in the light sensitivity of ChRs is reportedly closely correlated to their slow or prolonged kinetics (τoff). Based on this criterion, more sensitive mutants of ChR2 and CoChR have been developed by optimizing their off-kinetics [24,25]. Recently, using the bioinformatic approach, we identified ComV1 (ex3mV1Co), which is sensitive to dim light, via the modification of mVChR1 [21,27]. The photocurrent of ComV1 is approximately three times higher than that of mVChR1 at any wavelength. In addition, ComV1-expressing ganglion cells of a rat model of RP with degenerated photoreceptor cells produced a visual response to even 3.5 µW/mm^2^ of visible light. However, the τoff kinetics of ComV1 was slower than that of mVChR1. Mattis et al. reported that the τoff kinetics was negatively correlated with the photocurrent [26]. The time constants represented as τon and τoff, same as those for photocurrents, are important factors when ChRs are used for neural activation or silencing. In the visual system, the use of ChRs with longer time constants might affect flicker responses and produce a low-frame-rate vision.

In the present study, we compared the electrophysiological function of certain mutants of ComV1 to determine factors affecting the time constant and aimed to develop new ChRs with high light sensitivity and fast kinetics. Although attempts to improve the channel kinetics have been made [18,23,27,28], to our knowledge, this study is the first to investigate the factor that determines channel kinetics from the perspective of ChR structure.

## 2. Results

### 2.1. Photocurrent and Channel Kinetics of the H172R Mutant

It is known that the K176R mutation in Chrimson improves the photocurrent and off-kinetics [18]. The amino acid K176 in Chrimson corresponds to H172 of ComV1, which was identified as an important residue in the ion gating pathway, according to bioinformatic analysis (Figure 1a). Therefore, we selected H172 as a candidate replacement residue to improve the photocurrent and time constant of ComV1.

The expression of ComV1 and the H172R mutant In HEK293 cells resulted in Venus fluorescence, mainly observed on the plasma membrane (Figure 1b,c). The typical waveform of photocurrent is shown in Figure 1d,e. The photocurrents and wavelength sensitivities were not very different between ComV1 and the H172R-mutant (Figure 1f). Additionally, the decay at the peak wavelength did not change (Figure 1g). The τon for the H172R mutant was longer at certain wavelengths, but was not changed at the peak wavelength (Figure 1h). In contrast, the τoff of the H172R mutant was longer than that of ComV1 at 455, 505, 560 and 617 nm (Figure 1i).

### 2.2. Photocurrent and Channel Kinetics of the H172 Mutant

The τoff constant of the H172R mutant was significantly increased compared with that of H172, indicating H172 is the key amino acid for the channel kinetics. Although histidine and arginine are basic amino acids, the van der Waals radius of arginine is larger than that of histidine and the hydropathy of the two amino acids is different. Hence, the van der Waals radius or hydropathy of an amino acid at this site might affect the channel kinetics.

We created four additional mutants of H172G, H172A, H172K, and H172Y, which exhibited different properties, and transduced them into HEK293 cells to record the photocurrent. The typical waveforms are shown in Figure 2a. The τon kinetics of the H172A mutant was significantly shorter than that of ComV1 (Figure 2b). No significant difference was observed in the τoff between H172A and ComV1; however, the τoff of H172A was more than half of that of ComV1 (Figure 2c). These values did not change with the replacement of H172G. However, the H172K mutant showed significantly slower kinetics than ComV1; in particular, the τoff of the H172K mutant was 10 times slower than that of ComV1. The τoff of H172Y tended to be faster than that of ComV1; however, no significant difference was detected, and the τon of H172Y was equivalent to that of ComV1. Some of these results for channel kinetics are broadly consistent with the findings of recent studies on Chrimson [29] and ChR2 [30] that compared similar mutants.

The photocurrent of each mutant tended to be lower than that of ComV1 (Figure 2d). The photocurrent of H172A, H172K, and H172Y was significantly lower than that of ComV1 at 505 nm. The decay of H172G and H172A tended to be greater than that of ComV1, but without a significant difference (Figure 2e). They showed similar expression profiles as ComV1 (Figure 2f).

### 2.3. Analysis of the Correlation between Amino Acid Properties and Channel Function

Based on the results for the substitution of H172R, we hypothesized that the characteristics of amino acids affected the channel kinetics. We selected four amino acids with different characteristics and compared them with histidine for substitution at the 172nd position. The data showed that the van der Waals radius and solubility considered as the index of hydrophilicity (PubChem, NCBI, https://pubchem.ncbi.nlm.nih.gov/, accessed on 28 April 2021), of the 172nd amino acid affected the channel kinetics but not the induced photocurrent (Figure 3a–h). The van der Waals radius showed a weak positive correlation with the τon (Figure 3b, r = −0.3376, *p* < 0.01) and τoff (Figure 3c, r = 0.2861, *p* < 0.05). In contrast, the van der Waals radius showed a weak negative correlation with the decay (Figure 3d, r = −0.3361, *p* < 0.01). The solubility of the 172nd amino acid was consistent with the van der Waals radius (Figure 3f–h). In particular, the solubility showed a strong positive correlation with the τoff (Figure 3g, r = 0.9383, *p* < 0.001).

### 2.4. Structural Analysis with Molecular Dynamic Simulation

To analyze the structure of ComV1 and its mutants, molecular dynamic (MD) simulation was performed on the model at an early excited state (template PDBID: 7E71). The ion gating pathway constituted TM1, TM2, TM3, and TM7, and the inner gate of ComV1 comprised H172, E121, and R306 (Figure 4a). The major change induced by each mutation was the formation of an ion tunnel. The broadness of the bottleneck of the ion tunnel was not changed by H172G (Figure 4b); however, with H172A, the bottleneck expanded (Figure 4c). Conversely, with H172K and H172Y, the bottleneck narrowed (Figure 4d,e). In addition, we confirmed the difference in the amino acids forming the bottleneck between ComV1 and its mutants (Table 1). This difference could be due to the change in the interaction between the 172nd residue and the surrounding amino acids by their size or electric charge.

MD simulation showed that the mutation H172 changed the formation of the ion gate. The effect of various ion gates on the electrophysiological function of ChRs was verified using the correlation analysis. The photocurrent of the mutant, which has a wider ion gate, tends to be higher than that of ComV1 (Figure 5a). A previous study suggested that the widening ion gate causes an increase in ion influx [27]. However, the photocurrent of the H172A mutant was low even though it has a wider ion gate. Moreover, the channel kinetics (τon and τoff) and the decay correlate with ion gate broadness (Figure 5b,c). These results suggest that the amino acids forming the ion gate strongly influence the stability of the channel states.

## 3. Discussion

ComV1, which was previously developed by us [27], showed a higher sensitivity to light intensity in the order of µW/mm^2^ than mVChR1 [21]. However, the on- and off-kinetics of ComV1 were slower than that of mVChR1 [27]. Considering the application of ComV1 in retinal gene therapy, slow channel kinetics would negatively affect vision restoration. In the present study, we identified an amino acid residue related to the channel kinetics of ComV1 and demonstrated the correlation between amino acid properties and channel kinetics. Replacement of the single amino acid residue substantially improved the channel kinetics without affecting the photocurrent. This finding would be useful for further development of optogenetic genes for controlling cells by light and providing higher temporal and spatial resolutions.

The 172nd position in ComV1 was a part of the simulated ion gating pathway (Figure 1a). Previously, we showed that amino acid residues close to the ion gating pathway contribute to the photocurrent, as well as the channel kinetics. Indeed, ComV1 was developed by introducing amino acid sequences from other ChRs along with the ion gating pathway into mVChR1. According to a previous study, the mutation K176R in ChrimsonR along with the ion gating pathway considerably contributed to channel gating activity induced by an opsin shift [18,29]. Considering this result, the amino acid residue of ComV1 (H172) corresponding to K176R of Chrimson was additionally substituted to arginine from histidine. The H172R substitution in ComV1 affected the off-kinetics (Figure 1i) but not the photocurrent (Figure 1f). H172 residue of ComV1 interacted with the surrounding residues R306 and E121 (Figure 4a). We hypothesized that the channel kinetics could be altered if the condition of the interaction was changed by the substitution of H172 to another amino acid residue. Therefore, we created mutants in which H172 was substituted with various amino acids. We selected glycine, alanine, lysine, and tyrosine, which exhibit different properties, as candidates for H172 substitution. The channel kinetics, τon, and τoff were improved in mutants with H172A substitution (Figure 2b,c). In addition, the correlation analysis clearly indicated that the properties of amino acid affected the time constant (Figure 3). These results provide new insights into the modification of ChRs.

The correlation analysis showed that the hydrophobic 172nd residue enabled fast kinetics (Figure 3f,g). A previous study on C1C2 showed water entry into the cavity near H173 (H172 of ComV1) at the beginning of the channel opening [31,32]. In addition, closing of the channel follows the dewetting of helices [33]. Thus, the hydropathy of this residue could influence the process of channel hydration and change the condition of channel gating.

MD simulation indicated the variation in the ion gate formed near the 172nd residue between mutants. A recent structural analysis suggested that E122 of C1C2 (E121 of ComV1) is deprotonated in the ground state [34] and protonated in the process of photocycle [35]. This proton movement might change the formation of the ion gate via hydrogen bonds. Moreover, H172 and the surrounding amino acid residues can serve as proton donors or acceptors because of their electric charge. Considering the difference between each mutant with respect to the formation of the ion gate, this residue could determine the condition of the ion gate by changing the proton movement during the photocycle process. Indeed, the pKa of H172 calculated with DelPhiPKa ([36] (http://compbio.clemson.edu/pka_webserver/, accessed on 28 April 2021) was 6.99; therefore, even small structural changes can change the protonation state of H172. However, it is speculated that the mechanisms of the interactions and proton movement between H172 and surrounding residues do not depend much on pH because the K176H Chrimson mutant has low pH dependency [29]. The investigations of these mechanisms requires an open state structure.

According to a previous study, in ChR variants, mutation of residues around the retinal-binding pockets, such as in ChEF and ChIEF, affects the photocurrent and channel kinetics [28]. The previous study showed that ChIEF, in which I170V was introduced into ChEF, improved the channel closure rate, and reduced light sensitivity, providing an explanation for why the mutation destabilized the open state of the channel. This effect resulted in a higher energy requirement for channel opening and faster transition back to the nonconducting state after the removal of light. In the present study, the 172nd residue in ComV1 was close to the retinal-binding pockets. Compared with ComV1, ComV1 with H172A showed the same phenomenon of improvement in the channel closure rate (Figure 2c) with a reduction in light sensitivity (Figure 2d). Our result is supported by the finding of a previous study [27]. MD simulation indicated that the ion tunnel formed near the 172nd residue changes with the mutation of H172. In addition, the τoff was negatively correlated with the ion tunnel radius (Figure 5c), and the decay was positively correlated with the ion tunnel radius (Figure 5d). These results suggest that the expansion of the ion tunnel destabilizes the open state of the channel. Some studies have shown a correlation between light sensitivity and off-kinetics [24,25,26]. High sensitivity and slow kinetics may be due to the expansion of the ion tunnel. Additionally, a negative correlation was observed between the broadness of the ion tunnel and τon; however, this may be because the interaction of the 172nd residue with the surrounding residue was weakened. Nevertheless, the ion tunnel of H172A was wider even at ground state simulation (Appendix A). In summary, the properties of the amino acid at the 172nd position in ComV1 play an important role in channel gating. 

Gene therapy using a ChR with slow kinetics may result in a low-frame-rate vision. When using optogenetic-mediated gene therapy to target RGCs, an appropriate time constant, especially τoff, is likely needed for producing action potentials in RGCs [37]. Native ganglion cells spontaneously create spikes [38] via depolarization, responding to signals from the second order neurons, such as bipolar and amacrine cells. For example, an intense light generates a considerable number of spikes with a high frequency through the continuous depolarization of RGCs. Thus, the decay and τoff constant could be closely related to induce the action potentials. The optimum τoff needed to produce action potentials efficiently is not clear. ComV1 and the H172A mutant showed a similar photocurrent, and the H172 mutant exhibited fast kinetics. In future research, analysis of action potentials produced from RGC-transduced ComV1 or H172 mutants could generate important findings for optimizing the τoff constant.

In the present study, we identified one amino acid residue that determines the channel kinetics of ComV1, and subsequently developed a mutant with fast kinetics, H172A. The electrophysiological examination showed the correlation between the channel kinetics and amino acid properties at this site. MD simulation also showed the change in the ion gating pathway, such as interaction with the surrounding amino acid residues and the radius of ion gate. It was considered that these structural changes are a factor in determining channel functions. These results will be useful for developing more functional optogenetics gene. In addition, the H172A mutant may be a therapeutic tool for vision restoration in photoreceptor degenerative disease.

## 4. Materials and Methods

### 4.1. HEK293 Cell Preparation

HEK293 cells were maintained under 5% CO_2_ at 37 °C in minimum essential medium (Thermo Fisher Scientific, Tokyo, Japan) supplemented with 1% non-essential amino acids (Thermo Fisher Scientific), 1% antibiotic (Thermo Fisher Scientific), 1% Gluta-Max (Thermo Fisher Scientific), and 10% fetal bovine serum (Thermo Fisher Scientific). Passaging was performed with 0.02% ethylenediaminetetraacetic acid (Sigma-Aldrich, St. Louis, MO, USA) in Dulbecco’s phosphate buffered saline every three days.

### 4.2. Gene Transfection into HEK293 Cells

The plasmid vectors used for the transfection were constructed as previously described [39]. In brief, cDNA coding Venus fluorescent protein was fused in frame at the end of the ComV1-mutant cDNA. The ComV1 (GenBank acc. No. LC730505) plasmid vector contained a CAG promoter [40], woodchuck hepatitis virus posttranscriptional regulatory element [41], and human poly A signal. A point mutation of the 172nd histidine on ComV1 was created with a PCR-based method using the KOD-Plus-Mutagenesis kit (Toyobo, Osaka, Japan). The PCR conditions followed the protocol of the kit, and the sequences of the primers used are shown in Appendix A. The sequence of each mutant was confirmed using the Sanger-sequencing method (Genewiz, Kawaguchi, Japan).

Transfection of the plasmid vectors into HEK293 cells was performed using a previously described calcium phosphate-based protocol [27]. Two days before patch clamp recording, the calcium phosphate-DNA particles were added to the culture medium following the replacement of a fresh medium, and the culture medium was replaced after 6 h. The cells were cultured for 24–30 h and seeded on glass slides the day before the patch clamp recordings. Approximately 5–15% of HEK293 cells were transfected (Venus-positive) using this method.

### 4.3. Expression Profile of Each Gene Transfected into Cells

The expression of each gene transfected into HEK293 cells was observed as the fluorescence of Venus 1 day after transfection [27]. The transfected HEK293 cells were fixed on chamber slides and enclosed with DAPI fluoromount-G^®^ (Cosmo-Bio, Tokyo, Japan). The fluorescence of Venus and DAPI was observed using a laser microscope (Carl Zeiss, Tokyo, Japan).

### 4.4. Patch Clamp Recordings

Whole-cell patch clamp recordings on HEK293 cells expressing each gene were performed using the EPC-10 amplifier and Patch Master software (HEKA Electronic, Lambrecht, Germany) 2 days after the transfection. Photocurrents were evoked by stimulation using diodes emitting light (Mightex Systems, Pleasanton, CA, USA) of various wavelengths (405, 455, 505, 560, 617, and 656 nm) adjusted to 1 µW/mm^2^. The data were recorded with filtering at 10 kHz and sampling at 5 kHz. Recordings were performed in Tyrode’s solution containing 138 mM NaCl, 3 mM KCl, 1 mM CaCl_2_, 2 mM MgCl_2_, 4 mM NaOH, and 10 mM HEPES, adjusted to pH 7.4 with HCl. The internal solution contained 130 mM CsCl, 10 mM HEPES, 2 mM MgCl_2_, 0.1 M CaCl_2_, 10 mM NaCl, 2 mM Na_2_ATP, and 1.1 mM egtazic acid, adjusted to pH 7.2. The photocurrent was evaluated as the maximum value of the elicited current during light stimulation. The τon and τoff were determined from monoexponential fitting of the rise and decay of the photocurrent. τon is the time until the photocurrent reaches 1 − e^−1^ (63%) of the peak current during light stimulation (Appendix A). τoff is the time until the photocurrent decays to e^−1^ (37%) of the steady-state current after light stimulation. The decay was evaluated as the difference between the peak current and steady-state current divided by the peak current (Appendix A).

### 4.5. Statistics Analysis

All data in figures are expressed as mean ± standard error of mean. Statistical analyses were performed using GraphPad Prism ver. 4.00 software (GraphPad Software Inc., San Diego, CA, USA). The criterion for statistical significance was *p* < 0.05. All data were confirmed to be normally distributed with Kolmogorov–Smirnov test.

### 4.6. Molecular Dynamic Simulation

The first structural models of ComV1 and its mutants were built by homology modeling using the SWISS-MODEL server (https://swissmodel.expasy.org/, accessed on 22 March 2022). The template in the ground state structure is C1C2 (34, PDBID: 4YZI), and that in the early excited state is C1C2, 1 ms after light stimulation (32, PDBID: 7E71). The model consists of the ChR dimer, retinal, oleic acid, and water molecules.

MD simulation was run using the Discover/Insight II software package (Dassault Systemes BIOVIA, Discovery Studio Modelling Environment, Release, 2017, San Diego, CA, USA: Dassault Systemes, 2016). The consistent valence forcefield was used for protein and oleic acids, and extensible and systematic force field was used for retinal. While the structure was optimized, heavy atoms in proteins, water molecules, and oleic acids were fixed. The optimization was performed by 5000 steps for the minimization of conjugate gradient energy. An equilibration calculation was started from a force constant of 25–10 and 1 kcal/mol Å on heavy atoms of protein, oleic acids, and water molecules, respectively, and step-by-step equilibration multiphase system was run with 200-ps simulation. Subsequently, 1-ns simulation was performed under a constant pressure (NVT ensemble) at a constant temperature of 298 K.

## Figures and Tables

**Figure 1 ijms-24-05054-f001:**
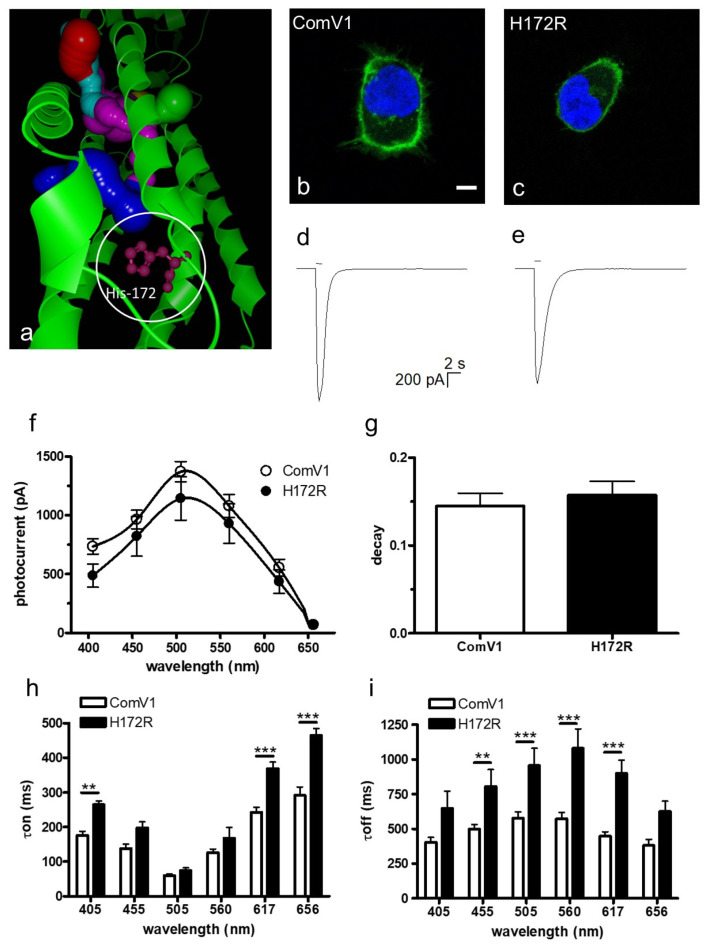
Comparison of functions between ComV1 and H172R. (**a**) The structure of ComV1 predicted using a bioinformatic tool. The protein is indicated by the green ribbon. H172 is located near the ion tunnel. (**b**,**c**) The fluorescence of 4′,6-diamidino-2-phenylindole (DAPI) (blue) and ComV1-Venus or H172R-Venus (green) introduced into HEK293 cells. The scale bar indicates 5 µm. Both genes are mainly expressed on the plasma membrane. (**d**,**e**) The typical waveforms of ComV1 (**d**) and H172R (**e**) were evoked with a 505-nm light-emitting diode adjusted to 1 µW/mm^2^ for 1 s. (**f**–**i**) Comparison of photocurrents (**f**), decays at 505 nm (**g**), τon (**h**), and τoff (**i**) between ComV1 and H172R. Data are expressed as mean ± standard error of mean (SEM) (ComV1 *n* = 18–19, H172R *n* = 8). The data were compared using two-way analysis of variance (ANOVA) with Bonferroni post-test (** *p* < 0.01, *** *p* < 0.001).

**Figure 2 ijms-24-05054-f002:**
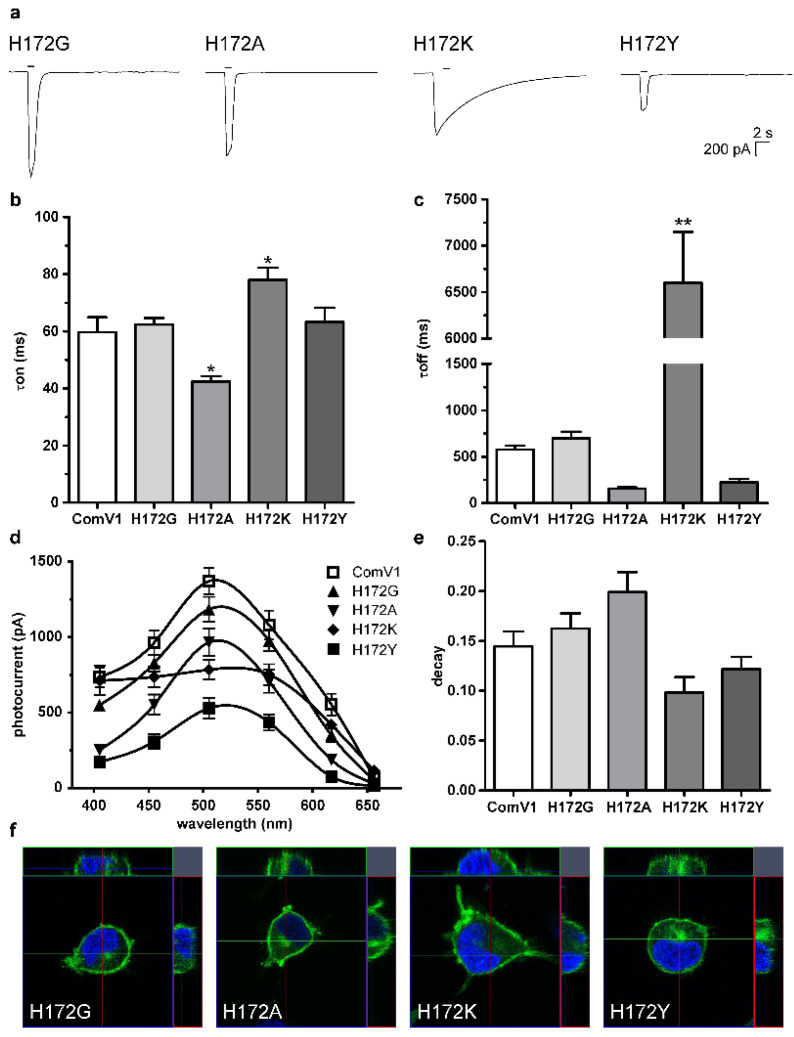
Electrophysiological profiles of ComV1 and the H172G, H172A, H172K, and H172Y mutants. (**a**) The typical waveforms evoked with 505-nm light adjusted to 1 µW/mm^2^ for 1 s. (**b**–**e**) Comparison of τon (**b**), τoff (**c**), photocurrents (**d**), and decays (**e**) between ComV1 and its mutants. τon, τoff, and decays at 505 nm (Appendix A). Photocurrents were evoked with a light-emitting diode adjusted to 1 µW/mm^2^ for 1 s. The data are shown as mean ± SEM (ComV1 *n* = 18–19, H172G *n* = 8, H172A *n* = 8, H172K *n* = 8, H172Y *n* = 8), and were analyzed using Dunnett’s multiple comparison test vs. ComV1 (* *p* < 0.05, ** *p* < 0.01). All data measured using light of various wavelengths and the results of statistical analysis are shown in Appendix A, respectively. (**f**) The fluorescence of channelrhodopsin-Venus (green) and DAPI (blue) in each mutant.

**Figure 3 ijms-24-05054-f003:**
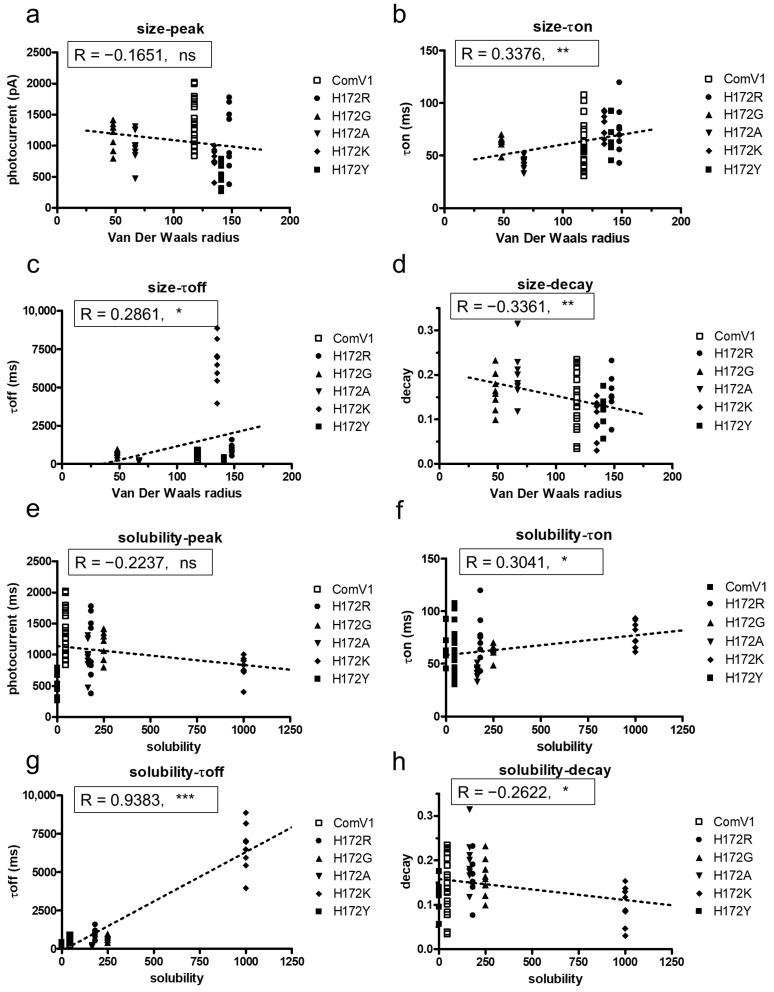
Correlation between properties of the 172nd amino acid and channel function of ComV1 and mutants. (**a**–**d**) Correlation between the size of the 172nd amino acid residue and channel properties, peak currents (**a**), τon (**b**), τoff (**c**), and decay (**d**) at 505 nm. (**e**–**h**) Correlation between the solubility of the 172nd amino acid residue and channel properties, peak current (**e**), τon (**f**), τoff (**g**), and decay (**h**). All data were analyzed with Pearson correlation coefficient (* *p* < 0.05, ** *p* < 0.01, *** *p* < 0.001, ns: not significant).

**Figure 4 ijms-24-05054-f004:**
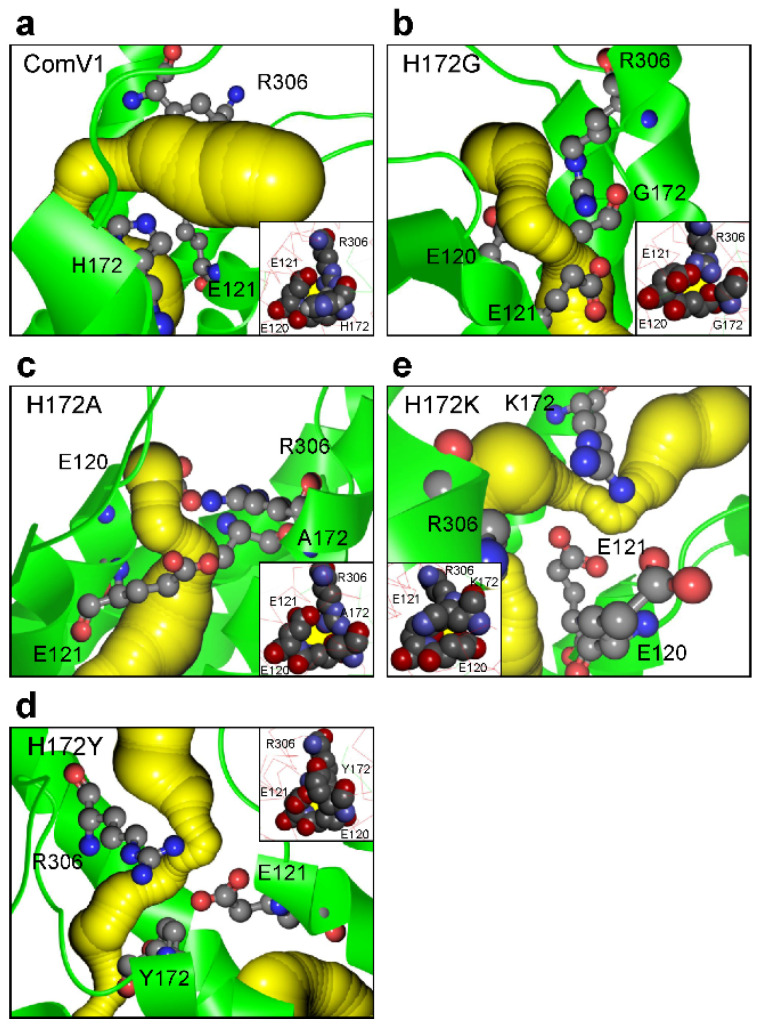
Structures of ComV1 and its mutants analyzed with molecular dynamic simulation. (**a**–**e**), The simulated structure around the 172nd residue in ComV1 (**a**), H172G (**b**), H172A (**c**), H172K (**d**), and H172Y (**e**) at an early excited state. The protein and the estimated ion gating pathway are shown as green and yellow ribbons, respectively. The figure in the frame is the structure near the 172nd residue observed vertically from the ion tunnel. The first structural models were built with homology modeling using the structure of PDBID: 7E71 as the template.

**Figure 5 ijms-24-05054-f005:**
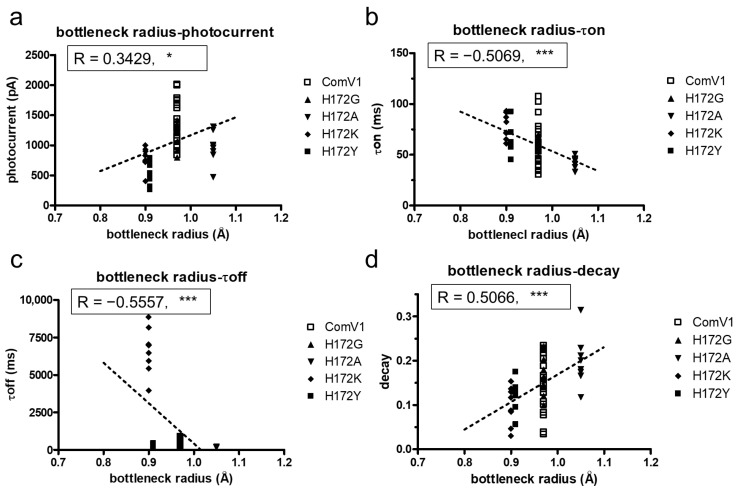
Correlation analysis between the ion tunnel radius and the channel properties. (**a**–**d**) Correlation between the radius of the bottleneck of the ion gating pathway and peak currents (**a**), τon (**b**), τoff (**c**), and decays (**d**) at 505 nm. All data were analyzed with Pearson correlation coefficient (* *p* < 0.05, *** *p* < 0.001).

**Table 1 ijms-24-05054-t001:** Properties of the ion gate bottleneck near the 172nd residue in the early excited state. The bottleneck radius and amino acids forming the bottleneck was calculated using molecular dynamic simulation.

	Bottleneck Radius (Å)	Amino Acids Residue Forming Bottleneck
ComV1	0.97	E121, H172, R306
H172G	0.97	E120, E121, (G172), R306
H172A	1.05	E120, E121, A172, R306
H172K	0.90	E120, E121, K172
H172Y	0.91	E120, E121, Y172

## Data Availability

The datasets used and/or analyzed during the current study are available from the corresponding author upon reasonable request.

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
