# Peer review of "Properties of a Single Amino Acid Residue in the Third Transmembrane Domain Determine the Kinetics of Ambient Light-Sensitive Channelrhodopsin"

_ijms, 2023, doi:10.3390/ijms24055054_

Round 1
Reviewer 1 Report
In this manuscript, Hatakeyama et al. demonstrated using channelrhodopsin mutants that a key amino acid, H172 regulates a light-induced ion gating and its kinetics (on and off rate) by the physicochemical properties of amino acid residues such as size, hydropathy and hydrophilicity etc. Furthermore, based on molecular dynamics, they could predict the on/off rate and extent of the gating elicited by the light. The authors conclude that several properties of H172, such as size, solubility, the interactions between H172 and the surrounding amino acid residues were involved in the ion gating. The proposed therapeutic strategy for photoreceptor degeneration disease, such as retinitis pigmentosa (RP) with mutant channelrhodopsin is promising, and pave the avenue to enable future design and development of functional channelrhodopsin to restore the lost function of photoreceptor. However, the determining factors for the gating rate and extent still remain unknown in detail. The readers would like to know the underling mechanism by which the ion gate is regulated.
Some specific comments are summarized below:
1. The authors highlighted the present mutant channelrhodopsins could be applicable to the discovery/design of mutant as potential mutant to compensate the lost function of photoreceptor. The readers would like to know exemplifications of compensation for the lost function of photoreceptor in PR.
2. What is the pKa value of H172 in the channelrhodopisn? Under acidic conditions, the residue of the histidine exists as the protonation form. Have the authors conducted the pH-dependency of the light-induced currents and the kinetics (on and off rate)? The authors would show the pH-dependency of photo-induced currents and discuss on the pKa value of H172 and the mechanism of the interactions among E120, E121, H172 and R306.
Minor comments
The authors used the term “solubility” of amino acid residue in this manuscript. What is the solubility in the protein? Does that term mean “hydrophilicity”?
Author Response
To reviewer 1,
Thank you for your detailed review and for giving us suggestions.
A kind thorough explanation is very helpful for us to revise our manuscript.
I revised our manuscript following the reviewer’s comments.
The revised manuscript received an English-editing service from a native speaker.
I appreciate it if you re-review our manuscript.
Sincerely,
Hiroshi Tomita,
Reviewer 1
- The authors highlighted the present mutant channelrhodopsins could be applicable to the discovery/design of mutant as potential mutant to compensate the lost function of photoreceptor. The readers would like to know exemplifications of compensation for the lost function of photoreceptor in PR.
RE: p. 1, lines 30-34 Added sentences into the introduction.
Several approaches have been investigated to compensate for the loss of function of photoreceptors, such as retinal prosthesis and stem cell transplant; however, useful therapeutic strategies have not been established. Recently, gene therapies using various optogenetic genes to target retinal ganglion cells (RGCs) have been developed for restoring vision in patients with RP.
- What is the pKa value of H172 in the channelrhodopisn? Under acidic conditions, the residue of the histidine exists as the protonation form. Have the authors conducted the pH-dependency of the light-induced currents and the kinetics (on and off rate)? The authors would show the pH-dependency of photo-induced currents and discuss on the pKa value of H172 and the mechanism of the interactions among E120, E121, H172 and R306.
RE: The pKa value of H172 was 6.99, calculated with DelPhiPKa. We haven’t conducted the pH-dependency. However, it is speculated that ComV1 has low pH-dependency because the kinetics of Chrimson K176H (corresponding H172 of ComV1) has low pH-dependency. In addition, H+ ion selectivity of ComV1 was lower than Chrimson, so all the more so.
Added the sentence into the discussion.
- 8, lines 242-247 Indeed, the pKa of H172 calculated with DelPhiPKa ([36] (http://compbio.clemson.edu/pka_webserver/) was 6.99; therefore, even small structural changes can change the protonation state of H172. However, it is speculated that the mechanisms of the interactions and proton movement between H172 and surrounding residues do not depend much on pH because the K176H Chrimson mutant has low pH dependency [29]. The investigations of these mechanisms requires an open state structure.
Minor comments
The authors used the term “solubility” of amino acid residue in this manuscript. What is the solubility in the protein? Does that term mean “hydrophilicity”?
RE: We considered “solubility” as the index of hydrophilicity. The values of solubility referenced PubChem (NCBI, https://pubchem.ncbi.nlm.nih.gov/). Modified sentence into the Results.
- 4, lines 139-142 The data showed that the van der Waals radius and solubility considered as the index of hydrophilicity (PubChem, NCBI, https://pubchem.ncbi.nlm.nih.gov/), of the 172nd amino acid affected the channel kinetics but not the induced photocurrent (Figure 3a–h).
Reviewer 2 Report
In their study, Hatakeyama and colleagues compared the electrophysiological function of certain mutants of the modified Volvox channelrhodopsin-1 (mVChR1), ComV1, with different amino acid residues at the 172nd position. The mutant, H172A, was characterized by fast kinetics, which prompted the authors to conclude that this variant might be useful for restoring vision in photoreceptor degenerative diseases.
Please find below my specific comments:
Introduction: The authors need to specify the ComV1 variant in more detail in the introduction. Which celllular structures in the retina are targeted by channel rhodopsins?
Results/Figure 1: Please check the figure and the figure legend. The graphs of τon, τoff, and decays are assigned to the wrong letters in the legend. In my opinion, the graphs of figures 1 f, h and i should be compared by repeated measures 2-way ANOVA and a posthoc test rather than just an unpaired t-test.
Methods: Some more details on the creation of the mutants, H172G, H172A, H172K, and H172Y, would be desirable.
Methods: What was the percentage of transfected (Venus-positive) HEK293 cells by using the calcium phosphate-based protocol? Why did the authors use the HEK cell line and why not a photoreceptor cell line? Also, the authors should add some sentences into the discussion on that issue.
Methods/Statistics: How did the authors determine whether their data were normally or non-normally distributed?
Author Response
To reviewer 2,
Thank you for your detailed review and for giving us suggestions.
A kind thorough explanation is very helpful for us to revise our manuscript.
I revised our manuscript following the reviewer’s comments.
The revised manuscript received an English-editing service from a native speaker.
I appreciate it if you re-review our manuscript.
Sincerely,
Hiroshi Tomita,
Reviewer 2
Introduction: The authors need to specify the ComV1 variant in more detail in the introduction. Which celllular structures in the retina are targeted by channel rhodopsins?
RE: The target cells of our gene therapy are the retinal ganglion cells. Added sentence into the introduction.
- 1, lines 32-34 Recently, gene therapies using various optogenetic genes to target retinal ganglion cells (RGCs) have been developed for restoring vision in patients with RP.
p.2, lines 60-62 In addition, ComV1-expressing ganglion cells of a rat model of RP with degenerated photoreceptor cells produced a visual response to even 3.5 µW/mm2 of visible light.
Results/Figure 1: Please check the figure and the figure legend. The graphs of τon, τoff, and decays are assigned to the wrong letters in the legend. In my opinion, the graphs of figures 1 f, h and i should be compared by repeated measures 2-way ANOVA and a posthoc test rather than just an unpaired t-test.
RE: Corrected figure 1 and legend. Values of Τon and τoff were analyzed with Two-way ANOVA with Bonferroni post-test.
- 2, lines 87-88 In contrast, the τoff of the H172R mutant was longer than that of ComV1 at 455, 505, 560 and 617 nm (Figure 1i).
- 3 lines 95-100 The typical waveforms of ComV1 (d) and H172R (e) were evoked with a 505-nm light-emitting diode adjusted to 1 µW/mm2 for 1 s. (f–i) Comparison of photocurrents (f), decays at 505 nm (g), τon (h), and τoff (i) between ComV1 and H172R. Data are expressed as mean ± standard error of mean (SEM) (ComV1 n = 18–19, H172R n = 8). The data were compared using two-way analysis of variance (ANOVA) with Bonferroni post-test (*p < 0.05, **p < 0.01, ***p < 0.001).
Methods: Some more details on the creation of the mutants, H172G, H172A, H172K, and H172Y, would be desirable.
RE: Added sentences and a table S4.
- 9, lines 305-308 A point mutation of the 172nd histidine on ComV1 was created with a PCR-based method using the KOD-Plus-Mutagenesis kit (Toyobo, Osaka, Japan). The PCR conditions followed the protocol of the kit, and the sequences of the primers used are shown in Table S4.
- 10, lines 373-374 Table 4S: The sequences of primers used PCR for producing mutants.
Methods: What was the percentage of transfected (Venus-positive) HEK293 cells by using the calcium phosphate-based protocol? Why did the authors use the HEK cell line and why not a photoreceptor cell line? Also, the authors should add some sentences into the discussion on that issue.
RE: Added sentence into the Methods
- 9, lines 315-16 Approximately 5–15% of HEK293 cells were transfected (Venus-positive) using this method.
In our gene therapy, the target cells are the retinal ganglion cells but not the photoreceptor cells. There are some reports about RGC-5 cells as the established retinal ganglion cell line. However, it is difficult to get the RGC-5. In addition, HEK293 cells are often used for channelrhodopsin-related studies. It is easy to compare the function of other channelrhodopsins published in others by using the HEK293 cells.
Methods/Statistics: How did the authors determine whether their data were normally or non-normally distributed?
RE: Determined with KS (Kolmogorov-Smirnov) test, and confirmed that all data were normally distributed. Added sentence into the Methods
p.10, lines 347-348 All data were confirmed to be normally distributed with Kolmogorov–Smirnov test.